# Impact of Daily Consumption of Whole-Grain Quinoa-Enriched Bread on Gut Microbiome in Males

**DOI:** 10.3390/nu14224888

**Published:** 2022-11-18

**Authors:** Liangkui Li, David Houghton, Georg Lietz, Anthony Watson, Christopher J. Stewart, Wendy Bal, Chris J. Seal

**Affiliations:** 1Human Nutrition Research Centre, Population Health Sciences Institute, Faculty of Medical Sciences, Newcastle University, Newcastle upon Tyne NE4 2HH, UK; 2Wellcome Centre for Mitochondrial Research, Translational and Clinical Research Institute, Newcastle University, Newcastle upon Tyne NE4 2HH, UK; 3Translational and Clinical Research Institute, Faculty of Medical Sciences, Newcastle University, Newcastle upon Tyne NE1 7RU, UK; 4School of Natural and Environmental Sciences, Newcastle University, Newcastle upon Tyne NE1 7RU, UK

**Keywords:** quinoa, gut microbiome, dietary fibre

## Abstract

Diets rich in whole grains are associated with improved health and a lower risk of non-communicable diseases, but the mechanisms through which these health benefits are conveyed are uncertain. One mechanism may be improvements in the gut environment by the delivery of fermentable substrates and associated phytochemicals to the lower gut and modification of the gut microbiome. Quinoa is included in the whole-grain category because of its structural similarities to cereals but the effects of its consumption on the gut microbiome have not been investigated to date. Our aim was to examine the impact of daily quinoa consumption on the gut microbiome in a 4-week randomised cross-over intervention separated by a 4-week wash-out period involving 28 adult males. Participants consumed either a quinoa-enriched wheat-bread roll providing 20 g quinoa flour each day, or a control wheat-only bread roll. Stool samples were collected in sterile collection tubes immediately before and at the end of each intervention period. DNA was then extracted, and the 16S rRNA V4 region of extracted DNA was amplified and sequenced. For both the control and quinoa bread periods, there were no changes at the phyla or genus level between baseline and week 4 (all *p* > 0.05). Diversity in the microbiome profile was not different from baseline after either intervention arms. The results show that small changes in the type of cereal consumed—substituting 20 g of refined wheat flour with whole-grain quinoa flour—was not able to significantly modulate the gut microbiome. Further studies with higher levels of quinoa or longer exposure periods are needed to ascertain if there is a dose–response effect of quinoa, and if these effects are able to translate into clinical outcomes.

## 1. Introduction

The higher consumption of whole-grain foods is widely reported to reduce the risk of chronic non-communicable diseases, including cardiometabolic diseases, type 2 diabetes, obesity and some cancers [1,2,3,4,5,6,7,8]. The mechanism(s) through which whole grains confer their associated health benefits remains unclear. However, many of the health benefits are likely to be due to the presence of dietary fiber and associated phytochemicals, a large part of which is lost during the refining process when the bran and germ are removed. Cereal fibres delivered through whole-grain foods include the full range of fiber types depending on the grain type, but include fibers characterised by their ability to hold water and their degree of fermentability including β-glucans, arabinoxylans, cellulose, and fructans [9]. 

Quinoa, part of the *Polygonaceae* family of plant species, is categorised as a whole grain; however, it is not a member of the *Graminaea* family of grasses, because the seeds have a similar nutritional composition to other cereals and the whole seed is consumed [10,11]. Quinoa contains a mixture of soluble (~1.3 g/100 g) and insoluble (~8.2 g/100 g) fiber, has a high protein content (~13.5 g/100 g), and provides carbohydrates that are digested slowly when compared to other cereals [12]. Interestingly, quinoa also contains high levels of free, conjugated and bound (poly)phenolics, particularly saponins [13,14], which have been associated with a number of health benefits [15]. The properties of quinoa contribute to its ability to pass through the upper gastrointestinal tract (GIT) and into the lower GIT, where it provides substrates for the gut microbiome, which undergo fermentation, producing products such as short-chain fatty acids (SCFA) and release bound (poly) phenolics and other phytochemicals, which become available for metabolism by the host [15].

The gut microbiome has attracted much attention over the last decade, primarily due to improvements in sequencing technologies, computational and molecular methods [16]. The gut microbiome functions through a complex symbiotic relationship with its host, contributing to digestion and absorption, specifically the fermentation of dietary fibre and other food components escaping digestion and absorption in the upper tract, providing a substrate for the growth in potentially beneficial bacteria [17], influencing health and disease [18]. Digestion and absorption in the gastrointestinal tract are both dependent on the composition of the gut microbiome [19]. Interestingly, the consumption of quinoa has been shown to modulate the gut microbiome in pre-clinical studies [20,21,22], and to possess prebiotic properties in vitro [23] and reduce blood glucose response when consumed in a bread matrix containing wheat flour, potentially due to the slower digestion [12]. Although quinoa may confer beneficial changes in the GIT, potentially due to changes in gut microbiome composition, further research is required to assess the effects of quinoa on the gut microbiome in human feeding studies. The primary aim of this analysis was to investigate changes in the gut microbiome profile following the consumption of quinoa-enriched bread in a randomised cross-over trial. 

## 2. Materials and Methods

The study was conducted in accordance with the Declaration of Helsinki, with all participants providing written, informed consent prior to the study commencement. Ethical approval was provided by the University of Newcastle Faculty of Science, Agriculture and Engineering Research Ethics Committee, reference 16-LI-034. The study was registered on ClinicalTrials.gov, registration number NCT03036618. All study procedures were conducted in the NU-Food Food and Consumer Research Facility at the University of Newcastle, UK.

Full details of the experimental protocol have been published elsewhere [12]. Briefly, the study was a randomized, controlled, cross-over trial consisting of two treatment periods consisting of 4 weeks, separated by a 4-week washout period. Participants were overweight but otherwise healthy males >35 years old, non-smokers, who were not taking medication [12]. During the treatment arm of the study, participants consumed one quinoa-enriched bread roll per day (approximate fresh weight 160 g providing 20 g quinoa flour). During the placebo arm of the study, participants consumed a control bread roll containing no quinoa (approximate fresh weight 160 g made with 100% refined wheat flour). Participants were asked to maintain their normal diets but were asked to avoid all other whole-grain foods during the study period, and they were provided with a list of foods to avoid. At the beginning and end of each 4-week treatment period, participants attended the research facility following an overnight fast (>8 h) and provided a fasted blood sample, and anthropometric measurements were taken (results reported in [12]. Participants provided a stool sample collected in sterile stool collection universal tubes immediately prior to the visit.

### 2.1. Stool Bacterial DNA Extraction and 16S rRNA Bacterial Profiling

Stool samples were stored at −80 °C until analysis for no more than 3 months. DNA was extracted from an approximately 300 mg stool using the FastDNATM Spin Kit for Soil (MP Biomedicals) following the manufacturer’s protocol. The 16S rRNA V4 region was selected for polymerase chain reaction (PCR) amplification, as previously described [24]. Sequencing was carried out in the Illumina MiSeq platform using the 2 × 250 bp paired-end protocol yielding pair-end reads that almost completely overlap. The primers applied in amplification possessed MiSeq sequencing and single-end barcodes, which allow for pooling and direct sequencing of PCR products [25]. Phylogenetic and alignment-based approaches were incorporated into the 16S rRNA gene pipeline data to maximize data resolution. The read pairs were demultiplexed based on the unique molecular barcodes, and reads were merged using USEARCH v7.0.1090, allowing zero mismatches and a minimum overlap of 50 bases [26]. Merged reads were trimmed at the first base with Q5. Additionally, a quality filter was used for the resulting merged reads, and reads were discarded if they contained above 0.05 expected errors. 16S rRNA gene sequences were clustered into Operational Taxonomic Units (OTUs), the term used to categorize groups of closely related bacteria at a 97% sequence similarity level by using the UPARSE algorithm [27]. OTUs were then mapped to an optimized version of the SILVA Database, containing only the region of 16S V4 to determine taxonomies [28]. Bacterial abundances were then mapped using the demultiplexed reads to the UPARSE OTUs. A custom script was then used to construct a rarefied OTU table using the output files generated for all analyses of alpha-diversity, beta-diversity, and phylogenetic trends [29].

### 2.2. Statistics

The statistical package R was used for the analysis and visualization of the gut microbiome communities. The phyloseq package was utilised to import, rarefy all samples to 5000 reads and calculate alpha- and beta-diversity metrics [30,31]. The Monte Carle permutations were utilised to estimate p-values for principle co-ordination plots, comparing the baseline against the end of each 4-week intervention. All p-values were adjusted, accounting for the total number of comparisons using a false-discovery (FDR) algorithm [32]. Data are reported as mean and standard deviations (SD ±) unless stated.

## 3. Results

### 3.1. Participant Characteristics and Fiber Intake

A total of 37 male participants were enrolled into the study. Nine dropped out during the study due to non-compliance with the study procedures. Twenty-eight participants completed the study with a mean age of 51.5 (±10.7) years, a body mass index (BMI) of 27.7 (±10.7) kg/m^2^ and a body fat percentage (%) of 25.4 (±5.2) at baseline. Systolic and diastolic blood pressures were 130 (±12) and 85 (±10) mmHg at baseline, respectively. Fasting glucose was 5.7 (±12) mmol/L. Participant’s BMI, body fat %, blood pressure and fasting blood glucose did not change at any time point during the study (*p* > 0.05) [12]. The quinoa bread contained higher levels of dietary fiber (6.52 g/100 g vs. 3.60 g/100 g for quinoa and control bread, respectively), especially insoluble fiber (4.99 g/100 g vs. 2.17 g/100 g). Total dietary fiber intake was not different between groups, either at baseline or at the end of the 4-week intervention (22.1 g/d vs. 20.1 g/d for control at baseline and end of intervention; 24.7 g/d vs. 22.5 g/d for quinoa bread at baseline and end of intervention; *p* for change = 0.687). The intake of other nutrients was not affected by treatment or duration of the intervention [12].

### 3.2. Bacterial Profiles

To investigate the impact of the quinoa and the control wheat breads on the gut microbiome, we compared stool samples collected prior to and following each of the 4-week arms of the intervention. The gut microbiome profiles in the current study reflect a composition expected from adult human stool samples, with bacteroidetes and firmicutes as the dominant phyla. In both the control and quinoa bread, there was a small but non-significant increase in the relative abundance of firmicutes, by 7.5 ± 15.2% vs. 4.4 ± 10.8%, respectively (both *p* > 0.05; Table 1) between baseline and week 4 of the intervention. In a similar manner, the relative abundance of bacteroidetes was reduced between baseline (−8.5 ± 15.4%) and week 4 (−5.4 ± 13.7%), although this did not reach statistical significance (*p* > 0.05; Table 1). In the control wheat bread group, there was a decrease in the relative abundance of the genera *Bacteroides* of −6.9% and an increase in *Fusicatenibacter* of 1.4% and *Subdoligranulum* of 1.9% when comparing baseline to week 4 (*p* > 0.05; Table 1). In the quinoa bread group, there were changes in the genera *Anaerostipes* and *Dorea* between baseline and week 4, but these failed to reach significance (*p* > 0.05; Table 1). However, none of the changes in taxonomic relative abundance reached statistical significance.

### 3.3. Alpha Diversity and Beta Diversity

α-Diversity was assessed by comparing the number of observational taxonomic units (OTU) before and after the two 4-week treatment periods. There were no significant changes in *α*-diversity between baseline and week 4 in either the control or quinoa bread (*p* > 0.05; Figure 1). In addition, there were no significant changes in weighted or unweighted UniFrac analysis between baseline and week 4 for the quinoa enriched bread (*p* = 0.52 and *p* = 0.21, respectively) or the control bread (*p* = 0.98 and *p* = 0.25, respectively) (Figure 2).

## 4. Discussion

To the best of our knowledge, this is the first study to investigate the effects of a quinoa-enriched food on diversity and composition of the gut microbiome in healthy male participants. We have demonstrated that 4 weeks of quinoa-enriched bread was unable to modify specific bacteria phyla and genera, nor could it modify the alpha- or beta-diversity of the gut microbiome.

Direct evidence for changes in bacterial composition due to quinoa consumption were previously reported only in pre-clinical animal models, where rats consuming a soluble polysaccharide extract from quinoa exhibited changes in the Firmicutes/Bacteroidetes ratio [20]. Garcia-Mazcorro et al. [22] compared the gut microbiome from obese mice fed either a quinoa-based diet (containing 83% quinoa) or a standard purified rodent diet, and obese mice fed the AIN-93G diet. The authors reported a significantly higher number of OTUs in the quinoa-fed obese mice compared with both lean and obese mice fed the control diet, but no significant impact on the firmicutes/bacteroidetes ratio. Pre-clinical data such as these provide evidence that dietary changes incorporating quinoa might modulate the gut microbiome in humans, providing a potential strategy to modulate unfavorable changes in the gut microbiome in metabolic disorders such as obesity. Dietary intake has been shown to rapidly and reproducibly modulate the gut microbiome; however, the modest changes in the current study suggest that the amount, type, or duration of quinoa used was not sufficient for microbiome modulation [33]. Notably, studies reporting large effects of dietary interventions are generally due to extreme changes in dietary intake, such as highly protein-based vs. highly plant-based [34]. 

The lack of difference in gut microbiome across all analyses in the current study are in accordance with previous human interventional studies, where patients increased their dietary fiber intake through the consumption of different types of whole grains, but showed no differences in diversity metrices [35,36,37,38]. In contrast, Martinez et al. [39] and Foerster et al. (2014) reported significant increase in alpha-diversity, following the consumption of whole grain-enriched products. Dietary fiber has repeatably been shown to influence the gut microbiome, primarily as it supplies substrates for bacterial growth during fermentation. The lack of consistency reported here and throughout the literature are confounded by the wide range of types of whole grain and doses used in the interventions [9]. Whole grains deliver a wide range of nutrients into the diet, some of which escape digestion in the upper intestine and can then be made available as substrates for the gut microbiome. These include dietary fibers ranging in the degree of solubility, fermentability and molecular size, which will have differential effects on the microbiome. The evidence surrounding quinoa indicates that it should be consumed as part of a healthy diet; however, whether the gut microbiome is involved as a mediator or not needs to be demonstrated in further work.

We did observe small but non-significant changes at the phyla level in both arms of the intervention, with an increase in the relative abundance of firmicutes, and a decrease in bacteroidetes, potentially due to similarities in fibre content. Differences in tehe amount consumed, solubility, fermentability and molecular size of dietary fibers all influence the gut microbiome, and may account for similar findings in the literature where wheat has been utilized as the main source of whole grains [36,40,41,42,43,44]. In contrast to our findings, Martínez et al. [39] reported that whole-grain barley, in isolation and when combined with brown rice, was able to significantly alter the relative abundance of Firmicutes and Bacteroidetes. Although the exact delivery of whole grains was different to the current study, the expectation would be that the dietary fiber would still be expected to reach the lower GIT and undergo fermentation, and, therefore, modulate the gut microbiome. The differences reported in the literature and in the present study may be due to the amount of dietary fibre consumed and/or the study population (e.g., European vs. North American) studies, where variability in gut microbiome composition is substantial both between and within participants [45].

The current study is not without limitations. While the sample size and duration of the current study are comparable with the previous literature, the between- and within-participant variability in the gut microbiome may have contributed to the lack of statistical significance in all analyses. Increasing the sample size would improve the power and allow for a more detailed analysis, adjusting for potentially confounding variables. In addition, although we showed no significant changes in the bacterial composition or diversity, we did not perform microbial gene expression or metabolic profiling, which would have provided a better evaluation of functional changes in the microbiome, so that conclusions regarding the effects of quinoa would be more robust. Furthermore, participants were all healthy males. Future studies should aim to include male and females across a broad age range, in case there are age and gender effects linked to quinoa consumption. The range of BMI in the current study may have impacted the results, given the impact of obesity on gastrointestinal health and the gut microbiome. Future studies may scrutinize the inclusion criteria to remove this potential confounder. The dosage of quinoa that participants were instructed to consume was relatively small when compared with previous studies, and further pre-clinical and human feeding studies are required to ascertain if there is a dose–response to quinoa.

In the present study we did not observe any significant changes in diversity metrices or at the phyla or genus level in either the quinoa or control wheat-bread arms of the study. This lack of response may have been due to the small dietary changes utilised in the current study. The dose of whole grain (at 16 g) is a small but achievable dietary change and is relevant to targets for whole-grain intake, but is lower than might be needed to effect health improvements [1,2,3,4,5,6,7,8]. Further work is required to assess if there is a dose–response relationship to quinoa consumption and changes in the gut microbiome, and what, if any, effect these changes in the gut microbiome composition may have on the GIT and overall health of the host.

## 5. Conclusions

To the best of our knowledge, this is the first study investigating the effect of quinoa consumption on the human gut microbiome. Whilst there were small changes in the gut bacteria, these failed to reach significance. Moving forward, further work is required to ascertain if there is a dose–response effect of quinoa on its ability to modulate the gut microbiome, including whole-genome sequencing and metabolic profiling, and if these changes can elicit physiological health benefits.

## Figures and Tables

**Figure 1 nutrients-14-04888-f001:**
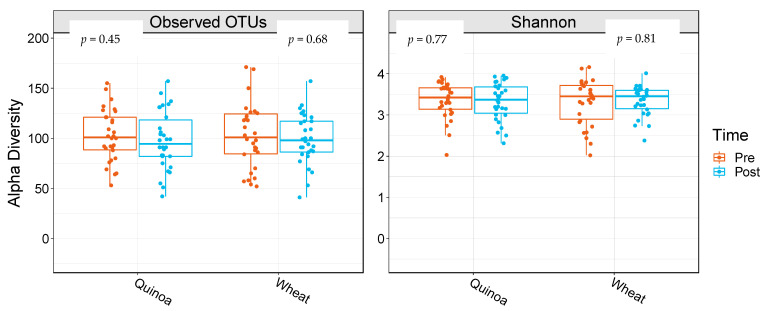
Boxplots showing alpha diversity differences for observed operational taxonomic units (OTUs) and Shannon Index pre- and post-quinoa and control wheat bread arms of the study. Data are presented as means and interquartile range (*n* = 28).

**Figure 2 nutrients-14-04888-f002:**
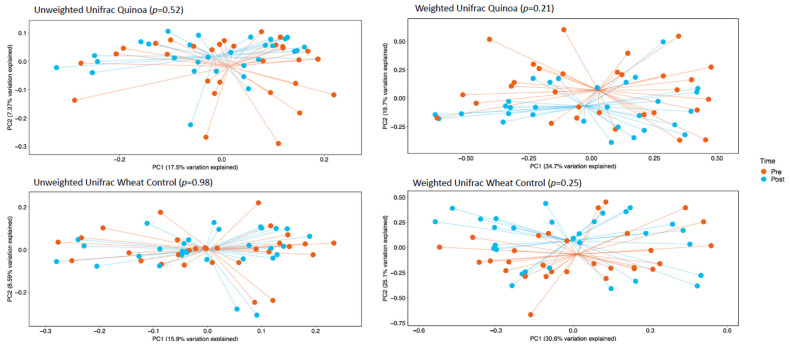
Principle co-ordinate analysis (PcoA) of unweighted UniFrac analysis and weighted UniFrac analysis pre- (baseline) and post-consumption (post) for quinoa and control wheat bread pre baseline and week 4.

**Table 1 nutrients-14-04888-t001:** Relative abundance of bacterial phylum and genus in stool samples of participants prior to and following 4 weeks consuming a control and quinoa bread roll (*n* = 28).

	Control Bread (*n* = 28)				Quinoa Bread (*n* = 28)					*p* (∆QR vs. ∆WR) ∆QR − ∆WR
	Baseline	Week 4	∆CB	*p* ^3^	Baseline	Week 4	∆QB	*p*	∆QB − ∆CB	
Actinobacteria	4.07 ± 3.57	5.31 ± 4.20	1.24 ± 3.65	0.40	5.07 ± 4.29	5.86 ± 4.83	0.79 ± 4.24	0.79	−0.45 ± 5.41	0.67
*Bifidobacterium*	2.97 ± 2.90	3.96 ± 3.49	0.99 ± 2.76	0.21	3.74 ± 3.63	4.23 ± 3.86	0.49 ± 3.67	0.49	−0.50 ±4.44	0.56
Bacteroidetes	25.01 ± 13.71	16.56 ± 11.26	−8.45 ± 15.39	0.21	23.35 ± 11.70	17.92 ± 12.22	−5.43 ± 13.65	0.11	3.02 ± 20.00	0.43
*Alistipes*	3.45 ± 3.35	2.21 ± 2.54	−1.24 ± 3.61	0.34	3.20 ± 3.20	2.23 ± 2.40	−0.96 ± 3.69	0.18	0.28 ± 2.72	0.60
*Bacteroides*	15.46 ± 13.79	8.57 ± 8.58	−6.89 ± 10.76	0.15	12.45 ± 9.29	10.34 ± 10.02	−2.10 ± 8.67	0.91	4.79 ± 13.70	0.08
Cyanobacteria	0.06 ± 0.15	0.02 ± 0.06	−0.04 ± 0.13	0.95	0.08 ± 0.30	0.09 ± 0.25	0.01 ± 0.39	0.78	0.05 ± 0.41	0.49
Euryarchaeota	1.67 ± 2.83	1.32 ± 2.19	−0.36 ± 1.71	0.93	1.11 ± 1.94	1.54 ± 2.20	0.44 ± 1.28	0.76	0.79 ± 1.57	0.05
Firmicutes	64.14 ± 14.51	71.65 ± 14.29	7.51 ± 15.16	0.30	65.55 ± 11.83	69.91 ± 12.33	4.36 ± 10.82	0.11	−3.15 ± 18.43	0.37
*Anaerostipes*	1.96 ± 2.05	2.64 ± 2.25	0.68 ± 1.87	0.70	1.70 ± 1.62	2.36 ± 1.93	0.66 ± 1.53	0.33	−0.02 ± 2.27	0.96
*Blautia*	3.43 ±2.48	4.70 ± 3.57	1.27 ± 3.53	0.72	3.92 ± 4.08	4.44 ± 3.20	0.52± 2.72	0.32	−0.75 ± 5.01	0.43
*Dorea*	1.87 ± 1.61	2.24 ± 1.37	0.38 ± 1.54	0.20	1.90 ± 1.36	2.50 ± 1.79	0.60 ± 1.50	0.40	0.22 ± 2.19	0.60
*Faecalibacterium*	7.57 ± 5.07	7.68 ± 5.72	0.11 ± 5.54	0.92	7.87 ± 5.18	7.56 ± 4.44	−0.31 ± 3.88	0.67	−0.42 ± 5.72	0.70
*Fusicatenibacter*	1.58 ± 1.44	3.01 ± 3.09	1.42 ± 2.76	0.19	2.21 ± 1.57	2.65 ± 2.39	0.45 ± 1.89	0.22	−0.98 ± 3.24	0.12
*Romboutsia*	5.06 ± 8.28	4.82 ± 5.74	−0.24 ± 7.02	0.86	4.41 ± 8.21	3.81 ± 4.00	−0.59 ± 6.45	0.63	−0.36 ± 5.85	0.75
*Subdoligranulum*	4.07 ± 3.01	5.97 ± 4.31	1.89 ± 3.47	0.44	5.25 ± 3.12	5.28 ± 3.87	0.03 ± 4.03	0.97	−1.86 ± 5.32	0.08
Proteobacteria	1.41 ± 1.20	2.05 ± 3.38	0.65 ± 2.85	0.61	2.50 ± 5.51	2.58 ± 5.53	0.08 ± 3.46 4646	0.77	−0.57 ± 4.28	0.49
Tenericutes	2.04 ± 5.29	1.30 ± 3.16	−0.74 ± 4.71	0.96	1.42 ± 3.68	0.89 ± 1.62	−0.52 ± 3.51	0.75	0.21 ± 6.06	0.85
Verrucomicrobia	1.51 ± 3.12	1.65 ± 2.81	0.14 ± 2.04	0.95	0.87 ± 1.70	1.18 ± 2.21	0.30 ± 1.25	0.78	0.17 ± 2.38	0.71

CB, control wheat bread; QB, quinoa-enriched bread, ∆, change between baseline and week 4, data presented as mean and standard deviation (SD ±).

## Data Availability

Data are not publicly available due to ethical constraints. However, anonymous patient data will be made available for all variables beginning 9 months and ending 36 months after the Article’s publication. All proposals for sharing data can be submitted up to 36 months following Article publication. Requests for data sharing should be addressed to the corresponding author (D.H.).

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
