# Peer review of "Impact of Daily Consumption of Whole-Grain Quinoa-Enriched Bread on Gut Microbiome in Males"

_nutrients, 2022, doi:10.3390/nu14224888_

Round 1

Reviewer 1 Report

The manuscript entitled “Impact of Daily Consumption of Whole-Grain Quinoa-Enriched Bread on Human Gut Microbiome”, in the manuscript authors have focused on quinoa-enriched bread to modulate gut microbiota in a randomized cross-over intervention. There are several serious concerns have been identified that lower the strength of work.

1. For the experimental design, I do not think that all the male participants were met the criteria of randomized study. Besides, the information of Materials and Methods provided is too brief, and it is hard to conclude the impact of quinoa consumption on gut microbiota. For example, the daily dietary intake may be one of the key points in modifying gut microbiota. In my opinion, the dietary composition and nutritional assessment should be incorporated in this study, at least the amounts of insoluble and soluble fiber intake per day in both 4-week period.

2. Increasing dietary fibers intake is well known for reduction of chronic diseases, such as type 2 diabetes, cardiovascular disease, and in gut health. I recommend that the introduction have to be re-written to complete the manuscript. The different physiological gastrointestinal responses of the insoluble and soluble fibers have been defined. Most soluble fibers are fermented by the gut bacteria; however, the insoluble fibers are not, or only slowly, digested by the gut bacteria. Thus, it is believed that the soluble fibers contribute to modify the colonization of gut microbial ecology. Due to the content of insoluble fiber in quinoa is 6 times higher than that of soluble fiber, it is relatively difficult to discover the impact of quinoa containing bread on microbiota.

3. In the discussion, the results did not be discussed in the broadest context. The previous studies mentioned in the discussion is not clear to explain the findings. For example, line 28 in the discussion page, the obese mice fed a quinoa-based diet (83% quinoa)…what does 83% quinoa means? Line 38-40, it is weird to mention high-protein-based and high plant-based diet? Line 43, what kinds of whole grains? Line 60, …potentially due to similarities in fiber content. Authors did not describe the daily fiber intake in the study.

Author Response

For the experimental design, I do not think that all the male participants were met the criteria of randomized study.

The study was carried out with male volunteers, and while there is no suggestion that the gut microbiome is directly affected by gender we accept that the results may not be applicable to all humans as implied in the title.  We have therefore changed the title to read “Impact of daily consumption of whole-grain quinoa-enriched bread on gut microbiota in males” and added a small expansion of the methods describing the participants.  Otherwise the study design was fully randomised as described in the paper.

Besides, the information of Materials and Methods provided is too brief, and it is hard to conclude the impact of quinoa consumption on gut microbiota. For example, the daily dietary intake may be one of the key points in modifying gut microbiota. In my opinion, the dietary composition and nutritional assessment should be incorporated in this study, at least the amounts of insoluble and soluble fiber intake per day in both 4-week period.

Full details of the experimental design, including dietary intake are included in the earlier paper published in this journal (Li, L., Lietz, G., Bal, W., Watson, A., Morfey, B., & Seal, C. (2018). Effects of quinoa (Chenopodium quinoa Willd.) consumption on markers of CVD risk. Nutrients, 10(6), 777.).  We believe that we have provided sufficient detail in the current paper in a way to avoid unnecessary replication, but have added specific detail relating to fiber intake since this is directly relevant to this paper as suggested by this, and the other reviewers (see below).

Increasing dietary fibers intake is well known for reduction of chronic diseases, such as type 2 diabetes, cardiovascular disease, and in gut health. I recommend that the introduction have to be re-written to complete the manuscript. The different physiological gastrointestinal responses of the insoluble and soluble fibers have been defined. Most soluble fibers are fermented by the gut bacteria; however, the insoluble fibers are not, or only slowly, digested by the gut bacteria. Thus, it is believed that the soluble fibers contribute to modify the colonization of gut microbial ecology. Due to the content of insoluble fiber in quinoa is 6 times higher than that of soluble fiber, it is relatively difficult to discover the impact of quinoa containing bread on microbiota.

We disagree that the introduction needs to be re-written with extensive description of dietary fiber digestion and metabolism which would make the introduction unnecessarily cumbersome.  We believe that key elements related to dietary fiber, quinoa and the purpose of this study are covered in the current Introduction.  Specific material relating to whole grains can be better found in the paper cited in the introduction (Seal, C. J., Courtin, C. M., Venema, K., & de Vries, J. (2021). Health benefits of whole grain:  effects on dietary carbohydrate quality, the gut microbiome and consequences of processing. Comprehensive Reviews in Food Science and Food Safety, 20(3), 2742-2768.). 

In the discussion, the results did not be discussed in the broadest context. The previous studies mentioned in the discussion is not clear to explain the findings. For example, line 28 in the discussion page, the obese mice fed a quinoa-based diet (83% quinoa)…what does 83% quinoa means?

We have suggested including ‘containing’ in the bracket so that the text now reads “…a quinoa-based diet (containing 83% quinoa) compared….” To make it clearer for the reader.

Line 38-40, it is weird to mention high-protein-based and high plant-based diet?

Our intention is to emphasise the fact that many studies show only small changes to the microbiome through dietary change, and it is only when the difference between diet types is extreme that differences are seen.  We believe that the current text is appropriate.

Line 43, what kinds of whole grains?

The studies cited all used different whole grains, so we have added this to the sentence which now reads “….consumption of different types of whole grains…”

Line 60, …potentially due to similarities in fiber content. Authors did not describe the daily fiber intake in the study.

We agree that quantity of DF consumed may affect the microbiome as well as the profile of fiber consumed.  We have added this to the sentence which mow reads “…Differences in amount consumed, solubility, fermentability and molecular…..”.  We have added comment in the paper about the amount of DF consumed in the present study.

Reviewer 2 Report

Authors report the effects of daily consumption of whole-grain quinoa-enriched bread on the human gut microbiome. The Introduction section provides all relevant data. Results are well presented, the Discussion is mainly successfully interpreted, and obtained results support Conclusions. The literature is relevant. The study design is explained in the previous article.

However, authors should provide more data on the nutritional characteristics of quinoa-enriched bread rolls and control bread rolls, primarily related to the dietary fibers and phenolic contents. Also, provide data on the estimated dietary fiber intake baseline and at the end of the intervention in both groups. Were there changes in the intake of nutrients before and at the end of the intervention study? Please add/mention these data in the Results and discuss them in the context of the effects of intervention studies with dietary fibers from other sources on the gut microbiome.

Author Response

Authors report the effects of daily consumption of whole-grain quinoa-enriched bread on the human gut microbiome. The Introduction section provides all relevant data. Results are well presented, the Discussion is mainly successfully interpreted, and obtained results support Conclusions. The literature is relevant. The study design is explained in the previous article.

We are grateful for the Reviewer’s supportive comments.

However, authors should provide more data on the nutritional characteristics of quinoa-enriched bread rolls and control bread rolls, primarily related to the dietary fibers and phenolic contents. Also, provide data on the estimated dietary fiber intake baseline and at the end of the intervention in both groups. Were there changes in the intake of nutrients before and at the end of the intervention study? Please add/mention these data in the Results and discuss them in the context of the effects of intervention studies with dietary fibers from other sources on the gut microbiome.

The proximate composition of the bread rolls and fiber intake during the study are reported in our earlier paper (Li et al., 2018) however, we have extracted relevant data and included this in the results section.  Additional comment has been included in the discussion as requested.

Reviewer 3 Report

 The author group did the first study investigating the effect of quinoa
consumption on the human gut microbiome. The idea is clear and the results are convincing . Here are some issues the authors may consider:

1. the body weight of the subjects varied a lot which might confound the results;

2. Only one dose was used. therefore, there are must strong evidice to suport that this dose is relevant to the daily intake of quinoa.

3. 16sDNA?

4. the figure 2 has some flaws, please fix.

5. in addition to the microbiota, any other data? like the short chain fatty acids?

Author Response

The author group did the first study investigating the effect of quinoa consumption on the human gut microbiome. The idea is clear and the results are convincing.

We are grateful for the Reviewer’s supportive comments.

The body weight of the subjects varied a lot which might confound the results.

We are not aware of evidence suggesting body weight is a driver for microbiome composition.  Since bodyweight was not affected by the intervention, the randomised cross-over nature of the study should account for any impact of body weight on the results.

Only one dose was used. therefore, there are must strong evidence to support that this dose is relevant to the daily intake of quinoa.

We acknowledge that using only one dose of quinoa in the study is a limitation in study design.  The amount of quinoa consumed at 16g is readily achievable using enriched bread as a medium of delivery.  This amount is, however, small compared with the amount recommended for improving health outcome.  A statement to this effect has been included in the summary to supplement the comment about the need to change the dose of quinoa consumed.

16sDNA?

This is correct but the sentence has been changed to read more clearly to “….and the 16S rRNA V4 region of extracted DNA was amplified….”

The figure 2 has some flaws, please fix.

The figure has been corrected.

In addition to the microbiota, any other data? like the short chain fatty acids?

Only a small fecal sample was collected sufficient for microbiome analysis so unfortunately no other measures were possible.